# Mapping the increased minimum mortality temperatures in the context of global climate change

Qian Yin [1], Jinfeng Wang [1,2,6]*, Zhoupeng Ren [1], Jie Li [3] & Yuming Guo [4,5,6]

Minimum mortality temperature (MMT) is an important indicator to assess the temperature–mortality relationship. It reflects human adaptability to local climate. The existing MMT estimates were usually based on case studies in data rich regions, and limited evidence about MMT was available at a global scale. It is still unclear what the most significant driver of MMT is and how MMT will change under global climate change. Here, by analysing MMTs in 420 locations covering six continents (Antarctica was excluded) in the world, we found that although the MMT changes geographically, it is very close to the local most frequent temperature (MFT) in the same period. The association between MFT and MMT is not changed when we adjust for latitude and study year. Based on the MFT~MMT association, we estimate and map the global distribution of MMTs in the present (2010s) and the future (2050s) for the first time.

[1] State Key Laboratory of Resources and Environmental Information System, Institute of Geographic Sciences and Natural Resources Research, Chinese Academy of Sciences. A11, Datun Road, Chaoyang District, Beijing, China. [2] University of Chinese Academy of Sciences, A19, Yuquan Road, Shijingshan District, Beijing, China. [3] School of Resources and Environment, Ningxia University, No.489, Helanshan Road, Xixia District, Yinchuan, Ningxia, China. [4] School of Public Health and Preventive Medicine, Monash University, Level 2, 553 St Kilda Road, Melbourne, VIC, Australia. [5] Department of Epidemiology and Biostatistics, College of Public Health, Zhengzhou University, No.100 Science Avenue, Zhengzhou, Henan, China. [6] These authors jointly supervised this work: Jinfeng Wang, Yuming Guo. *email: wangjf@lreis.ac.cn

Ambient temperature, an important environmental factor, affects human health in various ways. Many studies reveal that the associations between temperature and human mortality counts for non-external causes exhibit U-shaped, V-shaped or J-shaped curves[1–3], with the minimum mortality temperature (MMT) reflecting the most comfortable, optimum temperature. Mortality rates increase at temperatures outside the local MMT. Identifying the MMT has important practical implications for the investigation of the relationship between temperature and mortality, the definition of non-optimum temperatures, and the calculation of the mortality burden due to exposure.

More importantly, climate change is now widely recognized as the greatest global threat to human health in the 21st century[4]. MMT is very important for assessing the effects of temperature on human health under the changing climate. Although several researchers have predicted that global warming will lead to an increase in heat-related mortality and a decrease in cold-related mortality in the future[4], these studies have generally used an unchanging MMT as a reference value to estimate the relative risk of mortality in the future. Therefore, understanding the change in MMTs in the context of global climate change is important to assess the effects of temperature on mortality in the future.

However, MMT varies considerably across regions[1,3–6]. The existing MMT estimates are usually based on case studies in data-rich regions, and limited evidence about how to predict MMT is available at a global scale. Although many studies indicate that MMT strongly correlates with local annual mean temperature, latitude or some specific percentile of the local temperature (minimum mortality percentile [MMP])[3,5,7,8], it is still unclear what factors drive MMT and how MMT will change in the context of global climate change.

To fill the above research gap, we searched Web of Science, PubMed, Scopus and EMBASE for articles published in English from January 2000 to October 2018 using the terms temperature or heat, and death or mortality. Time-series regression or time-stratified case-crossover models combined with distributed lag non-linear model (DLNM)[1,3,9,10] are currently deemed as appropriate approaches to estimate temperature effects and provide comparable results. Thus, to ensure the results are comparable, we only investigate empirical MMTs that are modelled by DLNM. A total of 210 papers investigated the relationships between daily temperature and human mortality counts for non-external causes using the DLNM method. Excluding the literature from the repeated study areas, 16 papers[1,11–25] of which are chosen. These studies involved 420 locations from 30 countries, covering six continents (Antarctica was excluded).

Here, by analysing MMTs in these 420 locations, we found that although the MMT changes geographically, it is very close to the local MFT in the same period. Based on the MFT~MMT association, we estimate and map the global distribution of MMTs in the present (2010s) and the future (2050s) for each $0.5° × 0.5°$ grid.

## Results

### Spatially stratified heterogeneity of MMT. 
Figure 1 shows that MMT varies considerably across the planet. Among the 420 locations, the highest MMT reached 32 °C, while the lowest MMT was only 12 °C. When moving from low latitude to high latitude, MMT tends to decrease gradually. The mean MMP of the 420 locations is the 78th percentile, and 95% of the MMPs were within the 54th–92th range.

### Multiple linear regression (MLR) analysis. 
We used a multiple linear regression (MLR) to investigate the associations between MMTs and seven independent variables (Table 1).

Table 2 shows that Model 1 displays the lowest AIC and highest $R^2$. The adjusted $R^2$ of Model 1 is 0.86. In addition to this, we found that if we only use MFT ($x_1$) as predictor (Model 2), the predictive ability is as good as model 1 (the adjusted $R^2 = 0.84$). Model 2 performs much better than that of using annual mean temperature ($x_2$, adjusted $R^2 = 0.51$; Model 3), 78th percentile temperature ($x_3$, adjusted $R^2 = 0.60$; Model 4), and latitude ($x_4$, adjusted $R^2 = 0.28$; Model 5), respectively. The coefficients of Model 1 are shown in Supplementary Table 1. From Supplementary Table 1, we can see that the association between MFT and MMT is not changed when gross domestic product [GDP]/capita, latitude and the study year are adjusted. Therefore, to present a simplified model for practical use, we recommend using Model 2 to predict MMT.

### Driver of MMT-most frequent temperature (MFT). 
By analysing the MMT of the 420 locations, we found that although the MMT changes geographically, the MMT is very close to the local MFT of the entire year in the same period. We choose six representative cities to demonstrate our finding. (Supplementary Fig. 1 and Supplementary Table 2).

We compare the errors of using MFT, the 78th percentile temperature and annual mean temperature to represent MMT. The results (Fig. 2) show that the mean error between the MFT and MMT of the 420 locations is only ~0.9 °C (95%CI = 0–3 °C). If using MFT as the surrogate for MMT, the locations with an error of less than 1 °C, 2 °C, and 3 °C account for a proportion of 71%, 85 and 92%, respectively. This is much higher than using the 78th percentile temperature and the annual mean temperature to represent MMT, and the proportions of their errors within 1 °C are only 44 and 13%, respectively. These results show that MFT is a very good indicator of MMT. Supplementary Fig. 2 shows scatter plots of these three indicators and MMT. From Supplementary Fig. 2, we can see that MFT and MMT match best. Their fitting curves almost passes through the origin, with a slope of 1 and $R^2$ of 0.84 ($p < 0.001$). The results consistently show that the MFT is a good indicator of the MMT.

Table 3 shows the Pearson correlations ($r$ values, $n = 420$)) and $q$ statistics between three temperature indicators and the MMT. The MFT has a stronger correlation with the MMT than the other two mainstream indicators. The correlation coefficient between MFT and MMT reaches 0.93 ($p < 0.001$). The Geodetector $q$ statistic[26] reports that the MFT explains the 83% spatially stratified heterogeneity of the MMT, which is much higher than annual mean temperature (56%) and 78th percentile temperature (57%).

### Comparison of MMT and MFT in different climatic zones. 
The temperature characteristics and distributions of different climate regions have significant diversity. To test whether the relationship between MMT and MFT will change under different climatic conditions, we investigate the associations between the MFT, the 78th percentile temperature, the annual mean temperature, and the MMT in three main climate regions (tropical, subtropical, and temperate zones). The results are shown in (Fig. 3. From Fig. 3, we can see that in all climate regions, the MFT is the best indicator of MMT, particularly in tropical and temperate regions.

### The global distribution of MMT. 
Based on the MMT–MFT association, we can use MFT to estimate the MMT. Figure 4 shows the global distribution of the estimated MMTs during the 2010s for each $0.5° × 0.5°$ grid. The MMT varies considerably across the planet. The MMT tends to decrease gradually from low latitudes to high latitudes.

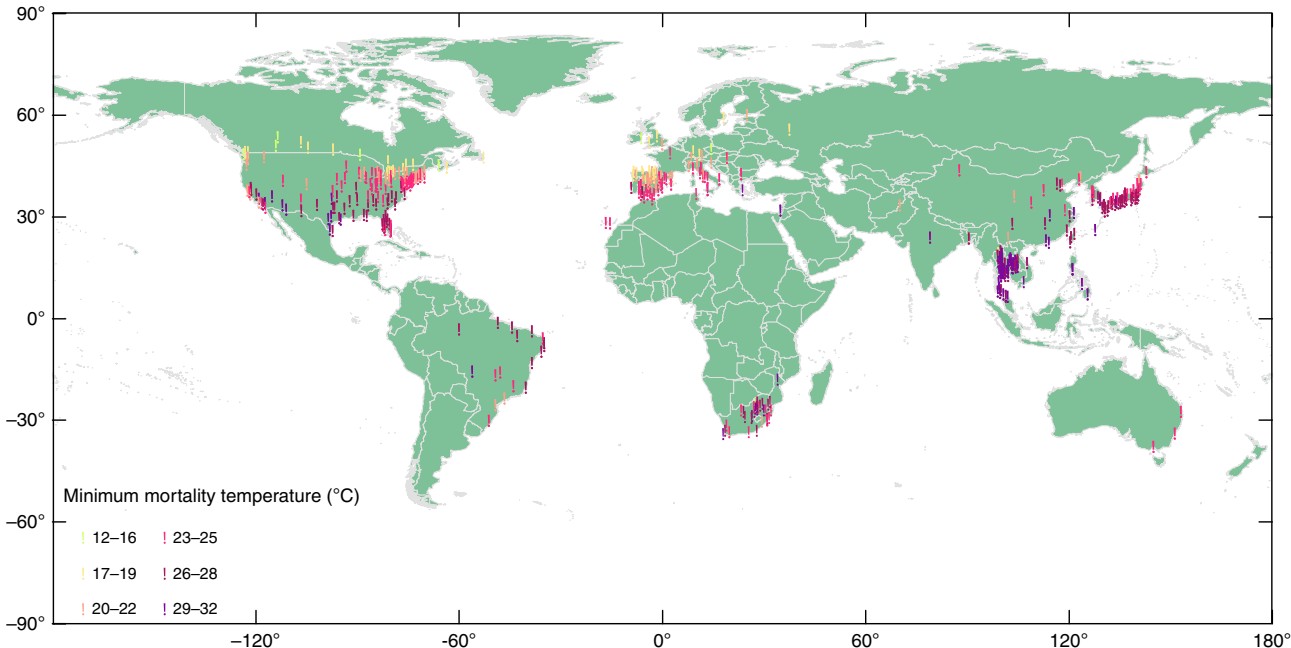

**Fig. 1** Overview of minimum mortality temperature (MMT) samples. MMTs in 420 locations are shown. From low latitude to high latitude, MMT tends to decrease gradually. Among these 420 locations, the highest and lowest MMTs are 32 °C and 12 °C, respectively

**Table 1 Independent variables considered in the statistical analysis**

| No. | Independent variables |
|---|---|
| $x_1$ | MFT (°C) |
| $x_2$ | Annual mean temperature (°C) |
| $x_3$ | 78th percentile temperature (°C) |
| $x_4$ | Latitude (°) |
| $x_5$ | GDP/capita (USD) |
| $x_6$ | Annual temperature range (°C) |
| $x_7$ | Study year (year) |

Dependent variable is MMT
Study year is the mean year of study period

**Future changes in MMT.** Based on the above conclusion—that the association between MFT and MMT is not changed when we adjust GDP/capita, annual mean temperature, latitude, and study year—we predict that with the changing climate, the future MFT can represent the future MMT. We proceed on the assumption that this prediction is correct, and then estimate the global distribution of MMTs in the present (2010s) and future (2050s) for each 0.5° × 0.5° grid. Figure 5 shows the distributions of changes of MFT in the 2050s compared to the 2010s under representative concentration pathway (RCP) scenarios RCP4.5 and RCP8.5. We can see that under the two scenarios, almost all land regions will experience increased MFT from the 2010s to 2050s. The grids with elevated MFT account for 94 and 96% of the total grids, with the global average MFT increasing by 1.1 °C (95%CI = 0–2.5 °C) and 1.8 °C (95%CI = 0.3–3.2 °C), respectively. The greatest increased values will occur in the mid- to high-latitude regions of the Northern Hemisphere, with the highest value reaching 4.5 °C. The changes are more rapid under the RCP8.5 scenario than the RCP4.5 scenario.

**Discussion**

The existing MMT estimates are usually based on case studies in data-rich regions, and limited evidence about MMT is available at a global scale, particularly in underdeveloped and developing areas. It is still unclear how MMT will change under global climate change. Our findings fill these research gaps. We found that although the MMT changes geographically, it can be well represented by the local MFT. Compared with two other indicators (annual mean temperature and 78th percentile temperature), MFT is the best indicator for fitting MMT. Furthermore, the association between MFT and MMT is not changed and the model performance is not changed when GDP/capita, latitude and the study year are adjusted (Table 2). The analysis of data from 420 locations provides evidence for this finding in a wide range of climates and populations with different demographic, socioeconomic, and infrastructural characteristics. Based on the MMT–MFT association, we have made the first estimate of the global distribution of MMTs in the present (2010s) and future (2050 s).

Our findings make an important contribution to the research on climate and human health. Humans adapt to climate in several ways, such as physiological, behavioural, and technological adaptations[8,27,28]. First, this study reveals that MFT is a good indicator that reflects an important aspect of how humans physiologically adapt to ambient temperature. Although the MMT changes geographically, it is significantly associated with the local MFT (Fig. 3). Darwin's theory of evolution states that: 'species have evolved principally via natural selection, and living forms evolved to improve them to better fit with the environment'. And human biological evolution is a process of pursuing self-advantage[29]. In the daily mean temperature histogram for a given year, MFT is the temperature that humans are most exposed to and, therefore, physiologically acclimatise to. Acclimatisation to the MFT may be considered as the pursuit of self-advantage maximisation in terms of temperature. In addition to physiological acclimatization, the socioeconomic conditions can affect human's behavioural and technological adaptations to ambient temperature. Some studies have confirmed the conclusion[24,30].

Second, to our knowledge, we have offered the first quantitative estimate of the global distribution of MMTs. MFT is easy to calculate, and we might, therefore, estimate the MMT in any

**Table 2 The model and performance of multiple linear regressions**

| Name | Model formula | Adjusted $R^2$ | AIC |
|---|---|---|---|
| Model 1 | MMT–MFT + Annual mean temperature + 78th percentile temperature + Latitude + GDP/capita + Annual temperature range + Study year | 0.86 | 1501 |
| Model 2 | MMT–MFT | 0.84 | 1531 |
| Model 3 | MMT– Annual mean temperature | 0.51 | 2005 |
| Model 4 | MMT–78th percentile temperature | 0.60 | 1906 |
| Model 5 | MMT–Latitude | 0.28 | 2168 |

*AIC akaike information criterion*

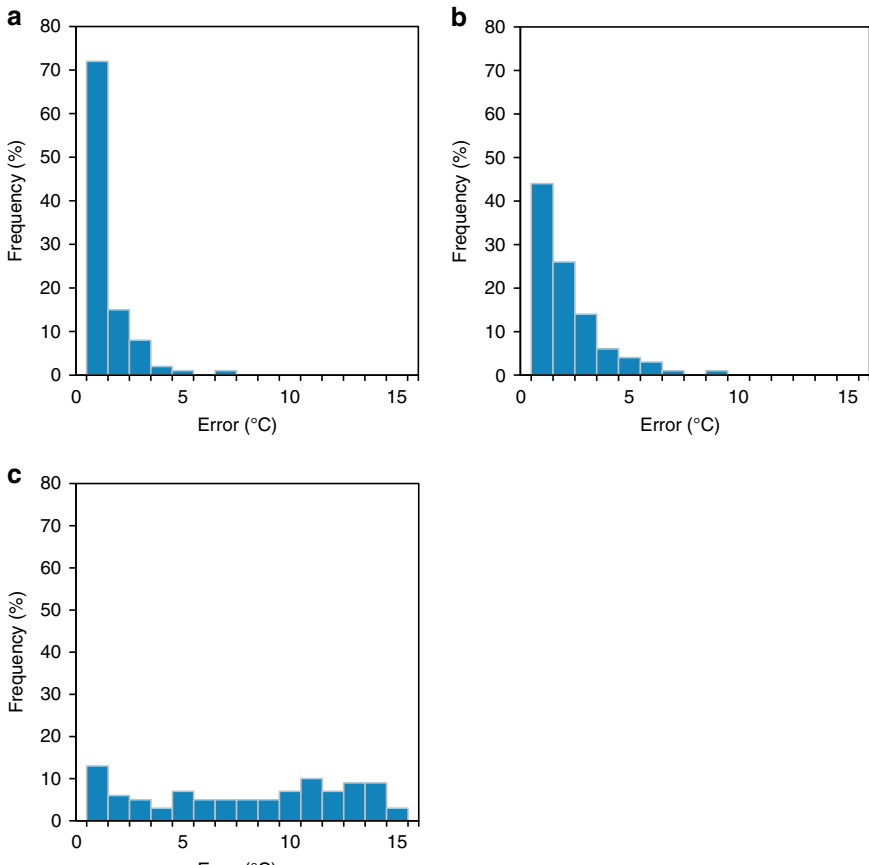

**Fig. 2** Error histograms for three temperature indicators as MMT. **a–c** Respectively show the error distributions of MFT as MMT, the 78th percentile temperature as MMT, and annual mean temperature as MMT. By comparing the errors of using three temperature indicators to represent MMT, MFT is proved to be a better indicator. **a** Using MFT as the surrogate for MMT, the locations with an error of less than 1 °C account for **a** proportion of 71%, which is much higher than using the 78th percentile temperature (44%, **b**) and the annual mean temperature (13%, **c**) to represent MMT

geographical location and in any period. Based on the current association between MMT and MFT, in order to validate whether the association is still reliable in the future, we have searched all available studies which provided the values of MMTs at different times in the same location. We found there were six papers which provided the MMT values of 62 locations from eight counties (USA[31], UK[31], Finland[31], France[9], Spain[32], Netherlands[33], Japan[34], and South Korean[35]) in different years. We calculated the MFTs of these locations during the same period. We found that: (1) In most locations, the temporal trends of MMT and MFT are consistent[9,31–35]; (2) The MMT and MFT have increased in most locations of the studied countries[9,31–35]; (3) Although in some locations MMT did not change or reduced, MFT had the similar trend in most of these locations[32,35]. For example, Donaldson found that from 1972 to 1997, MMTs rose 3.6 °C in North Carolina, 2.7 °C in Southeast

England and 1.3 °C in South Finland[30], and the MFTs in these three locations during the same period have increased 3 °C, 2 °C, and 1 °C, respectively (Supplementary Fig. 3). Similar results have also been reported in other countries such as Japan[34] and Spain[32]. Chung found that from 1972 to 2012, MMT rose about 4.8 °C in Japan, and the MFT in the same period has increased 5 °C. Achebak found that from 1980 to 2016, MMT rose from 19.5 °C to 20.3 °C in Japan, and the MFT in the same period rose from 19.5 °C to 20.5 °C. Based on this finding, we estimate the global distribution of MMTs in the future (Fig. 5).

Third, in the context of global warming, MMT may increase in the future[9,30–34] (Fig. 5). The existing prediction that global warming will lead to an increase in heat-related mortality and a decrease in cold-related mortality is based on unchanged MMT or MMP, may be questionable[36–38]. Instead of using unchanging MMT, future MMT predicted by future MFT might

**Table 3 Statistical indices between the three temperature indicators and MMT**

|  | Annual mean temperature | 78th percentile temperature | MFT |
|---|---|---|---|
| Pearson correlation | 0.71 | 0.75 | 0.93 |
| $q$ statistics | 0.56 | 0.57 | 0.83 |

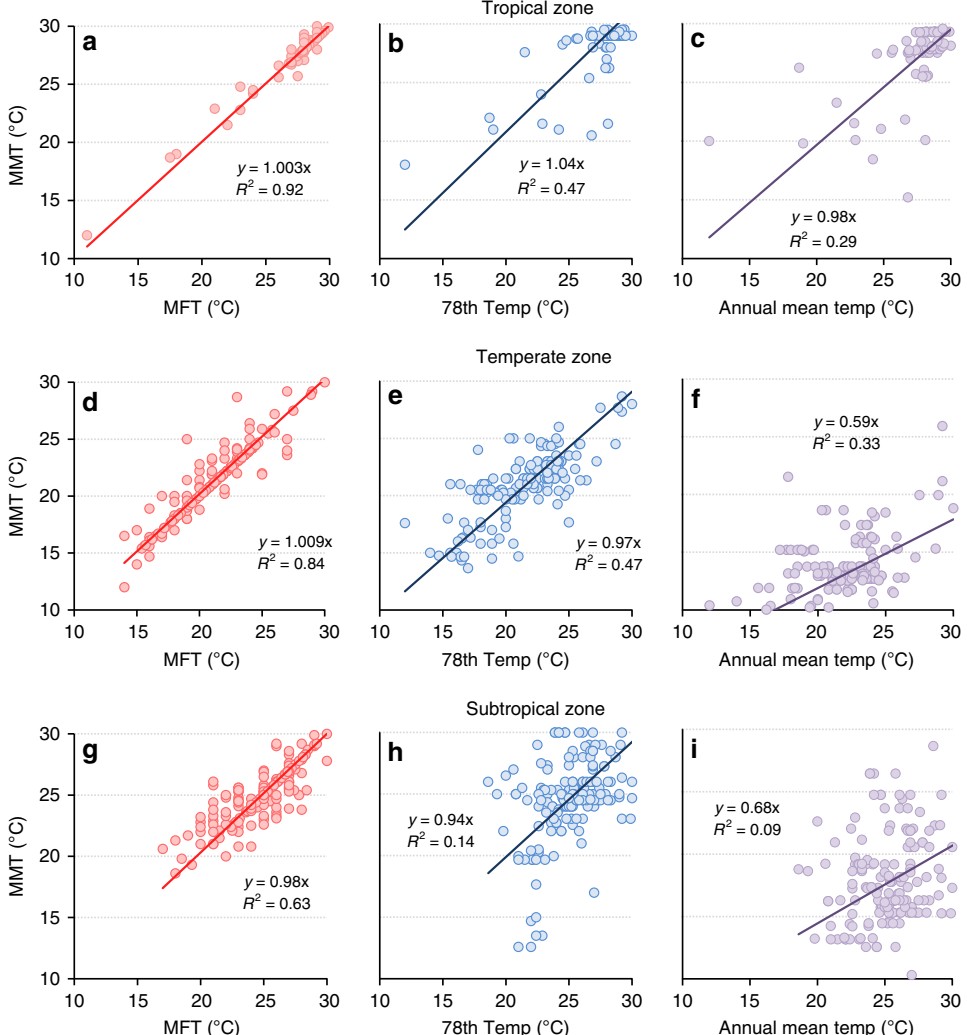

**Fig. 3** Correlations of MFT, the 78th percentile temperature, annual mean temperature and MMT for three climatic zones. **a–c** for tropical zone. **d–f** for temperate zone. **g–i** for subtropical zone. The fitting lines and performances are shown in the figures. MFT is proved to be the best indicator of MMT, particularly in tropical (**a**) and temperate regions (**d**)

provide more precise predictions on the mortality risks/burden due to ambient temperatures.

Different climate regions have diverse climate characteristics and temperature distributions, which may lead to different regular patterns of MMT. For three main climate zones (tropical, subtropical, and temperate region), we compared the associations between the MFT, the 78th percentile temperature, the annual mean temperature, and the MMT. The results showed that in all climate regions, the MFT is the best indicator of MMT, particularly in tropical and temperate regions.

This study has some limitations. First, similar to all global analyses, our data was not available for every country. Our study regions did not cover the tropical desert, plateau mountain, and cold climate regions. Notwithstanding, because these three missing regions have very small resident populations, our

research results could provide a reference for the vast majority of populations in the world. Second, the collected MMTs of the 420 locations were obtained from different studies, and the effect factors and parameter specifications considered in their models are not all the same. We acknowledge that the different parameter specifications could cause uncertainties. However, many studies conducted sensitivity analyses to check the robustness of their findings, and the analyses indicate that, the results were only slightly different for the different parameter specifications. Third, based on the association between MMT and MFT in the present, and the changing trend of MMT over time, we estimate the MMTs in the future. However, whether the future MFT can represent the future MMT is dependent upon the speed with which MMT tracks the changing MFT. There are still uncertainties regarding the speed of human adaptation to

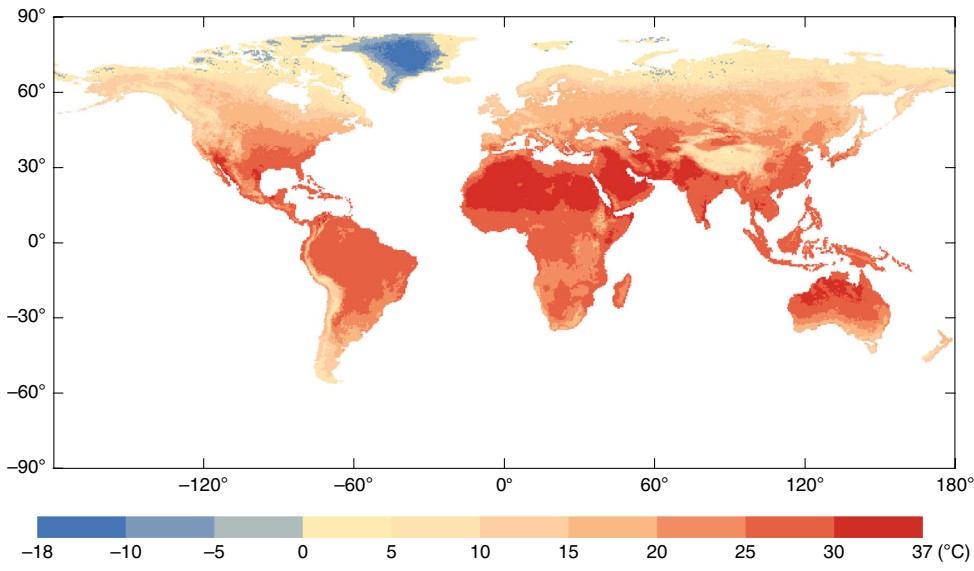

**Fig. 4** The global distribution of minimum mortality temperatures estimated for each 0.5° × 0.5° grid during the 2010s

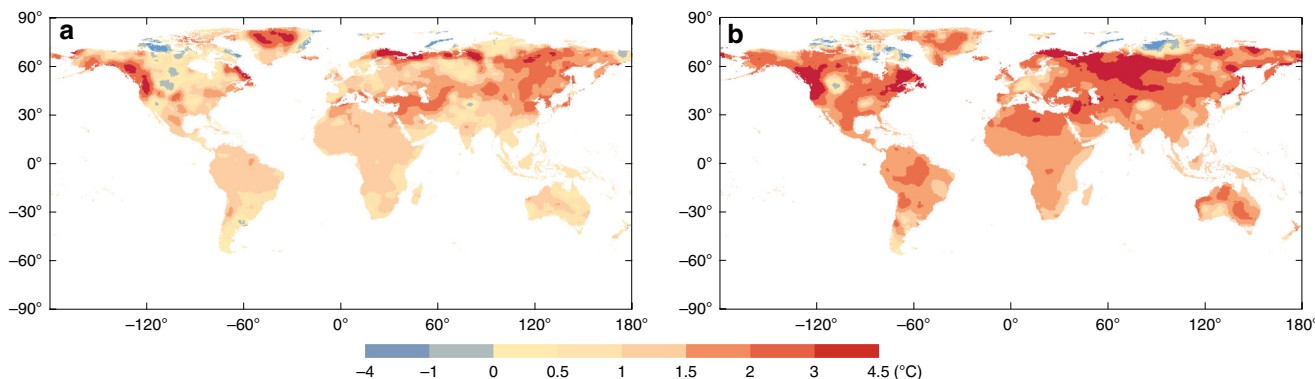

**Fig. 5** The distributions of change of most frequent temperature (MFT) in the 2050s compared to the 2010s under RCP4.5 (**a**) and RCP8.5 (**b**) scenarios for each 0.5° × 0.5° grid

temperature in the future. Nevertheless, this limitation exists in all studies on future projections, because we all don't know what will happen in the future. And our method provides a possible future scenario to improve the projection for future temperature–mortality associations.

**Current temperature**. A daily time series of the mean temperature of 420 locations during the study period was downloaded from the United States National Oceanic and Atmospheric Administration's (NOAA) website (https://www.ncdc.noaa.gov/).

**Future temperature**. The global projected daily mean temperatures from two RCPs (RCP4.5 and RCP8.5) under the five general circulation models (NorESM1-M, MIROC-ESM-CHEM, IPSL-CM5A-LR, HadGEM2-ES, and GFDL-ESM2M) during the 2010s and 2050s were obtained from the Inter-Sectoral Impact Model Inter-comparison Project (ISI-MIP)[39]. To be more suitable for use in climate change impacts studies, this dataset was bias corrected to reduce the biases in the original simulated temperatures, and it was statistically downscaled to improve the spatial resolution[40].

**Gross domestic product (GDP)**. GDP/capita data was employed at the national level in 2012, which was obtained from World Bank Open Data.

## Methods
The MFT of each location was obtained by calculating the mode of daily mean temperature distributed in the 54th–92th range during the entire year, which is the 95% distribution of the MMPs. We systematically analysed the distribution of the MMTs, annual mean temperature, 78th percentile temperature and MFTs in 420 locations from 30 countries. We compared the two existing indicators (annual mean temperature and 78th percentile temperature) and the new indicator (MFT).

We applied a MLR model to fit the MMT according to the seven independent variables. The model is defined below (Equation (1)):

$$y = \alpha + \sum_{i=1}^{7} \beta_i x_i \qquad (1)$$

where $y$ is the MMT, $\alpha$ is the intercept. $x_1, \dots x_i$ ($i = 1,\dots,7$) are the independent variables in Table 1, and $\beta_1, \dots \beta_i$ ($i = 1,\dots, 7$) are the regression coefficients.

We mapped the error (Equation (2)) histograms of the above three temperature indexes and calculated the Pearson correlations among them and the MMT.

$$Error = |T - MMT| \qquad (2)$$

where $T$ refers to the three temperatures mentioned above.

The 420 locations in the present study covered eight of the 11 major climatic zones, (including tropical grassland, tropical monsoon, tropical rainforest, subtropical monsoon, temperate continental, temperate maritime, temperate monsoon, and Mediterranean climates, excluding for the tropical desert, plateau

mountain, and cold climates). We combined eight climatic zones into three categories: temperate climate (including 91 locations), subtropical climate (including 174 locations), and tropical climate (including 155 locations) regions. For these three main climate regions, we analysed the associations between the MFT, the 78th percentile temperature, the annual mean temperature, and the MMT (Fig. 3).

With the projected daily mean temperature under five different global-scale general circulation models (GCMs) and two Representative Concentration Pathways scenarios (RCP4.5 and RCP8.5), we estimate the global distribution of MMTs in the present (2010s) and future (2050s) for each $0.5° × 0.5°$ grid (Figs. 4 and 5).

**Reporting summary**. Further information on research design is available in the Nature Research Reporting Summary linked to this article.

## Data availability

The authors declare that all data supporting the findings were obtained from open data. Data sources are available from the corresponding author upon request.

## Code availability

Code sources are available from the corresponding author upon request.

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

## Acknowledgements

This work was supported by the National Natural Science Foundation of China (grant nos: 41531179 and 41421001) and the Ministry of Science and Technology of China (grant nos: 2014FY121100 and 2016YFC1302504). Yuming Guo was supported by the Career Development Fellowship of the Australian National Health and Medical Research Council (APP1107107 and APP1163693). The funders played no role in determining the study design, data collection, or analysis methods employed, in our decision to publish, or in preparing the paper.

## Author contributions

Qian Yin, Jinfeng Wang, and Yuming Guo conceived of and designed the study. Qian Yin and Zhoupeng Ren collected and analysed the data. Qian Yin carried out the computations and wrote the paper. Qian Yin, Jinfeng Wang, Yuming Guo, Zhoupeng Ren, and Jie Li contributed to the final version of this paper.

## Competing interests

The authors declare no competing interests.

## Additional information

**Supplementary information** is availabel for this paper at https://doi.org/10.1038/s41467-019-12663-y.

