## [Peer Review File · Nature Communications]

Reviewers' comments:

Reviewer #1 (Remarks to the Author):

MAJOR POINTS

This paper presents a very interesting thesis: that "MMT is always identical to MFT". If true (or even nearly true), this has important implications for the projection of mortality under climate change this century.

However, I think you should be more measured in your choice of words. Figure 3 shows R-squared as low as .62 – that is a long way from "always identical". Throughout, I think the paper would benefit from more cautious presentation.

In the Abstract you write: "Moreover, MMT will increase in the future under climate change". The use of the word 'moreover' suggests that you intend to claim this as a separate finding from your discovery that at present, MMT tends to be close to local MFT. But you present no evidence for this major additional conclusion. Perhaps you really meant "therefore", but it does not follow automatically. Either way, you need to argue, with evidence, why this might be so.

Essentially you assume that because current MMT equals current MFT, under climate change this identity continue to hold. That is not a safe assumption: it depends critically on the speed with which MMT may track changing MFT. On page 5 you write "with the climate changing gradually, we expect that the future MFT can represent the future MMT". Similarly on page 6: "with the climate changing gradually, it is reasonable to assume that the future MFT can represent the future MMT". This is not justified or reasonable. Human populations have had thousands of years, at least, to adapt to their local conditions. Climate change is occurring over decades: not gradual at all compared to the ecological timescale of our species. You acknowledge this in the Abstract: "organisms fit to local temperature after a long-term adaptation", but then you seem to forget it.

It seems to me that merely acknowledging this as a limitation of the study (and at present you do not even do this) will not be sufficient. It is too central to your claim about projecting mortality.

MINOR POINTS

'in situ' is inappropriate where you just mean local. E.g. for "the MFT in situ" I suggest "the local MFT".

Abstract: "MFT consists of the adaption (Most) and term (Frequency) of temperature". I have no idea what this means.

Page 2: "The mean MMP ... 78th (95% CI: 54th, 92th)" This seems implausibly wide as a CI for the mean percentile from a sample of 423 locations. Perhaps you have calculated $\pm 2SD$ instead of $\pm 2SE$? If you mean that 95% of the MMPs were within this range (which does seem plausible), that is not a confidence interval.

Page 2 line -2: "A more common alternative..." I am not sure what you mean by "common".

Page 2 Table 1: please delete the p-value indicators (***) , they are irrelevant here. They represent tests that the correlations are zero, which is of no interest. Also in the text. However you do present a comparison of interest: "MFT explains [83% of the] spatial[ly] stratified heterogeneity of the MMT, much higher than the others". It would be appropriate (subject to journal policy) to give a p-value for that comparison.

Page 3 Figure 2: I suggest using a common X-axis scale (0-15, say) for a, b and c, to facilitate comparison between the three measures.

Page 6 line 6: "Based on this finding, to our knowledge, we estimated ...". I don't know what you mean by "to our knowledge" here: perhaps delete it.

Page 6 line 9: "this study reveals how humans adapt to ambient temperature". This claims too much, you have thrown light on just one aspect of this complex subject.

Page 6 line 11: "MFT accurately reflects the temperature that local people are most frequently exposed [to] and therefore adapt to". What about indoors? – most people surely do not live their lives exposed to the outdoor temperature.

Reviewer #2 (Remarks to the Author):

The authors investigated what indicators are associated with MMT (minimum mortality temperature) in global scale. They found that MFT (most frequent temperature) is strongly associated with MMT, particularly in tropical and temperate zone. This is very attractive finding, which will provide many implications in public health studies on climate change. They also presented one implication that future MFT (based on temperature projection scenarios) can represent future MMT. I appreciate the authors' hard work. However, the manuscript and analysis should be improved to support the finding and projection. My three main comments are below.

1. On ability to acclimate to the MFT.

I think that a fundamental mechanism to support that MFT can represent MMT is "People have the ability to acclimate to the local temperature following their long-term exposure to the MFT" (Line 149-150)

To claim this, the authors cited a paper "Population and Environment, Vol 33, 233-242". However, this paper is about an application of a modeling and I could not find out any relevance, in order to elucidate mechanisms behind why human is able to acclimate to the MFT. I think that the authors should thoroughly discuss mechanisms.

2. On independence to the other socioeconomic conditions

The authors claims that "it is identical to the most frequent temperature (MFT) in situ, independent to the other socioeconomic conditions" (Line 17-18). Unfortunately, I could not find any evidence to support this claim. I hazard a guess that the reason is that they found that strong correlation between MFT and MMT (with small errors) over cities and countries with different socioeconomic conditions. However, this does not necessarily indicate so. Does the authors argue that adaptation to MMT is acquired without any socioeconomic conditions, only with physiological acclimatization? In other words, without adaptation through socioeconomic conditions, does the authors argue that population can adapt to temperature to the same extent? Please clarify.

3. On temporal variation of MMT and predictability of this variation in future

The authors claimed that they "can estimate the MMT in any geographical location [in any period.]" (Line 24-25)

However, to predict in time dimension is not identical to predict in space dimension. The manuscript showed only spatial variation of MMT estimated from empirical data (unfortunately, I could not find what statistical methods they used to estimate MMT in the manuscript). The manuscript did not include [empirical evidence on temporal variation of MMT in the past]. Of course, I agree that MMT may vary globally over time, but I could not find the evidence in References #1-8 either (cited in the paragraph starting with "However, MMT varies considerably across regions". Whether MMT can vary over time must be empirically demonstrated first. Without this demonstration, how can the authors predict MMT in future?

Here is another issue. As suggested by the result of subtropical region, MFT is not as a strong

predictor of MMT as is in the other regions. The authors stated "In subtropical region, matching is not as good as in the other two regions, partly due to the peak value of temperature distribution in some subtropical regions are not obvious. When the temperature distribution is relatively even throughout the year, people's adaptation to temperature is not very obvious."(without any citation) (Line 171-174). To me, this result indicates very important limitations to predict MMT in the future. In future, temperature distribution may also change over time and climate zones may shift over time. Is it reasonable to assume that MFT can represent MMT in future, without investigating future change of temperature distributions in regions and climate zones? I think no.

Reviewer #3 (Remarks to the Author):

This is an original and thoughtful contribution to the field of heat-health impact assessment in the context of a changing climate.

One major issue that is not addressed and which is central to the interpretation of this paper, is that of how the minimum mortality temperature (MMT) was computed in the studies that are included in their quantitative analysis. The authors seem to view the MMT as some sort of fundamental quantity of nature, when in fact it is just a summary metric that emerges from a particular statistical model. If the model were to change, the MMT might change. Thus it's important to add a section defining MMT as used here and describing how it was estimated in the included papers. Do the author's conclusions only apply to one particular MMT computation method? i.e., Were papers only included that used one method for deriving the MMT? That might explain why other important papers that have reported "MMTs" seem to be missing. Todd and Valeron EHP 2015 is one prominent example. Also, many of the China heat-mortality published by Tiantian Li and colleagues, e.g., Li et al., Scientific Reports 2016.

Minor comments:

Line 22: Adaption is not a word.

Line 51: remove ? after MMT

Line 56: What were the exclusion criteria? Many papers that "investigated the relationships between temperature and human mortality counts for non-external causes" seem to be missing. So there must have been other exclusion criteria (e.g., "had to report MMT using method X").

Line 136: replace "identical to" with "well approximated by"

Line 142: delete "to our knowledge"

Line 162: include other citations that have questioned this, including:

Kinney et al., Environmental Research Letters 2015; Ebi and Mills, Climatic Change 2013; Staddon et al., Nature Climate Change 2014.

Reviewers' comments:

Reviewer #1 (Remarks to the Author):

MAJOR POINTS

Comment A1:

This paper presents a very interesting thesis: that “MMT is always identical to MFT”. If true (or even nearly true), this has important implications for the projection of mortality under climate change this century.

However, I think you should be more measured in your choice of words. Figure 3 shows R-squared as low as 0.62 – that is a long way from “always identical”. Throughout, I think the paper would benefit from more cautious presentation.

Response A1: We thank the reviewer very much for the insightful comments and suggestions. We have replaced “always identical” by “well represented by” (See Line 18). We also have carefully checked presentation throughout the paper.

Comment A2:

In the Abstract you write: ““Moreover, MMT will increase in the future under climate change”“. The use of the word ‘moreover’ suggests that you intend to claim this as a separate finding from your discovery that at present, MMT tends to be close to local MFT. But you present no evidence for this major additional conclusion. Perhaps you really meant “therefore”, but it does not follow automatically. Either way, you need to argue, with evidence, why this might be so.

Response A2: We have improved the statement as “We found that although the MMT changes geographically, it is well represented by the local most frequent temperature (MFT). The association between MFT and MMT was not modified by socioeconomic conditions and period. Because MFTs will increase in the future, we predicted MMT will increase accordingly.” (See Lines 17-23).

Comment A3:

Essentially you assume that because current MMT equals current MFT, under climate change this identity continue to hold. That is not a safe assumption: it depends critically on the speed with which MMT may track changing MFT. On page 5 you write “with the climate changing gradually, we expect that the future MFT can represent the future MMT”. Similarly on page 6: “with the climate changing gradually, it is reasonable to assume that the future MFT can represent the future MMT”. This is not justified or reasonable. Human populations have had thousands of years, at least, to adapt to their local conditions. Climate change is occurring over decades: not gradual at all compared to the ecological timescale of our species. You acknowledge this in the Abstract: “organisms fit to local temperature after a long-term adaptation”, but then you seem to forget it.

It seems to me that merely acknowledging this as a limitation of the study (and at present you do not even do this) will not be sufficient. It is too central to your claim about projecting mortality.

Response A3: We thank the reviewer very much for the insightful comments. We have improved the statements in the revision (See lines 210-224).

In addition, we have added relevant evidence in the discussion, as following: “Several previous studies have shown that MMTs changed over time¹⁻⁴. For example, Donaldson found that MMTs rose consistently with the increase of annual mean temperature from 1971 to 1997, by 3.6°C in North Carolina and by 2.7°C in Southeast England and by 1.3°C in South Finland¹. Ekamper found that during the past 150 years, the MMT increased from slightly below 15°C to around 17°C in the Netherlands². Similar results have also been reported in other countries such as the United States and Europe. By analyzing the temperature-death relationships in 52 Spanish cities, Tobías found that MMTs rose almost exactly at the same rate as annual mean temperature (1°C/°C). This suggests that Spanish had broadly adapted to their local climate to the same extent⁴.”

“Acknowledging the matching between MMT and climate change, we estimate the global distribution of MMTs in the future. We found that, under RCP4.5 and RCP8.5 scenarios, compared to the 2010s, in the 2050s the global average MMT increase by 1.1°C (95%CI=0-2.5°C) and 1.8°C (95%CI=0.3-3.2°C) respectively (Figure 4).”

However, whether the future MFT can represent the future MMT is dependent upon the speed with which MMT may track changing MFT. There are still uncertainties for the speed of human adaptation of temperature in the future. (See lines 261-264)

MINOR POINTS

Comment A4:

'in situ' is inappropriate where you just mean local. E.g. for "the MFT in situ" I suggest "the local MFT".

Response A4: Done (See Line 20).

Comment A5:

Abstract: "MFT consists of the adaption (Most) and term (Frequency) of temperature". I have no idea what this means.

Response A5: We have deleted this sentence.

Comment A6:

Page 2: "The mean MMP ... 78th (95% CI: 54th, 92th)" This seems implausibly wide as a CI for the mean percentile from a sample of 420 locations. Perhaps you have calculated $\pm 2SD$ instead of $\pm 2SE$? If you mean that 95% of the MMPs were within this range (which does seem plausible), that is not a confidence interval.

Response A6: We have improved the statement as "The mean MMP of the 420 cities was 78th, 95% of the MMPs were within 54th-92th range." (See Line 80).

Comment A7:

Page 2 line -2: "A more common alternative..." I am not sure what you mean by "common".

Response A7: "common" is not appropriate, we have deleted this sentence.

Comment A8:

Page 2 Table 1: please delete the p-value indicators (***), they are irrelevant here. They represent tests that the correlations are zero, which is of no interest. Also in the text. However you do present a comparison of interest: "'MFT explains [83% of the] spatial[ly] stratified heterogeneity of the MMT, much higher than the others'". It would be appropriate (subject to journal policy) to give a p-value for that comparison.

Response A8: We have deleted the p-value indicators. We have added the indicator (q statistics) comparison of spatially stratified heterogeneity. MFT explains the 83% spatially stratified heterogeneity of the MMT, much higher than annual mean temperature (56%) and 78th quartile temperature (57%). Pearson correlations and q statistics index were calculated independently. They both suggested that MFT is better than the other two indicators (See lines 119-122).

Comment A9:

Page 3 Figure 2: I suggest using a common X-axis scale (0-15, say) for a, b and c, to facilitate comparison between the three measures.

Response A9: Done (See Line 113 and Figure 2).

Fig. 2 | The error histograms by MFT (a), 78th quartile temperature (b), and annual mean temperature (c) as the MMT, respectively.

Comment A10:

Page 6 line 6: “Based on this finding, to our knowledge, we estimated ...”. I don’t know what you mean by “to our knowledge” here: perhaps delete it.

Response A10: We have deleted “to our knowledge” from the sentence (See Line 191).

Comment A11:

Page 6 line 9: “this study reveals how humans adapt to ambient temperature”. This claims too much, you have thrown light on just one aspect of this complex subject.

Response A11: We have improved the statement as “This study might reveal an important aspect of how humans adapt to ambient temperature.” (See Line 195).

Comment A12:

Page 6 line 11: “MFT accurately reflects the temperature that local people are most frequently exposed [to] and therefore adapt to”. What about indoors? – most people surely do not live their lives exposed to the outdoor temperature.

Response A12: We thank the reviewer very much for considering indoor temperature. However, this paper is specifically focusing on outdoor temperature. We cannot provide conclusions on indoor temperatures.

Reviewer #2 (Remarks to the Author):

The authors investigated what indicators are associated with MMT(minimum mortality temperature) in global scale. They found that MFT(most frequent temperature) is strongly associated with MMT, particularly in tropical and temperate zone. This is very attractive finding, which will provide many implications in public health studies on climate change. They also presented one implication that future MFT (based on temperature projection scenarios) can represent future MMT. I appreciate the authors’ hard work. However, the manuscript and analysis should be improved to support the finding and projection. My three main comments are below.

Response: We thank the reviewer very much for the encouraging comments. We

have revised the paper according to your comments and suggestions as following.

Comment B1:

1. On ability to acclimate to the MFT.

I think that a fundamental mechanism to support that MFT can represent MMT is ““People have the ability to acclimate to the local temperature following their long-term exposure to the MFT”” (Line 149-150)

To claim this, the authors cited a paper ““Population and Environment, Vol 33, 233-242””. However, this paper is about an application of a modeling and I could not find out any relevance, in order to elucidate mechanisms behind why human is able to acclimate to the MFT. I think that the authors should thoroughly discuss mechanisms.

Response B1: We have deleted the reference in revision. We have improved the statement and discussed the potential mechanisms in the revision (See lines 199-204).

By analyzing the MMTs of 420 locations, we found that although the MMT changes geographically, the MMT is significantly associated with the local MFT.

Our findings might reveal an important aspect of how humans adapt to ambient temperature. Darwin’s theory of evolution states that: “species have evolved principally via natural selection, and living forms evolved to improve them to better fit with the environment”. And human biological evolution is a process of pursuing self-advantage⁵. MFT is the longest period of climate histogram which human expose to and therefore acclimate to. Acclimatisation to the MFT can be considered to pursue self-advantage maximization.

Comment B2:

2. On independence to the other socioeconomic conditions

The authors claims that “it is identical to the most frequent temperature (MFT) in situ, independent to the other socioeconomic conditions” (Line 17-18). Unfortunately, I could not find any evidence to support this claim. I hazard a guess that the reason is that they found that strong correlation between MFT and MMT (with small errors) over cities and countries with different socioeconomic conditions. However, this does not necessarily indicate so. Does the authors argue that adaptation to MMT is acquired without any socioeconomic conditions, only with physiological acclimatization? In other words, without adaptation through socioeconomic conditions, does the authors argue that population can adapt to temperature to the same extent? Please clarify.

Response B2: We thank the reviewer for the insightful comments. We are sorry that we did not make a precise expression in the last version. We believe that socioeconomic conditions affect human adaptation to local temperature. However, we found the association between MFT and MMT is slightly modified by socioeconomic conditions. We mean that MFT is a better indicator than other two mainstream indicators for MMT and the association is independent to the socio-economic conditions. We have added more information in the revision as:

We applied a standard multiple linear regression model (MLR) to fit the MMT according to the independent variables. Model is defined as follows (Eq.(2)):

$$y = \alpha + \sum_{i=1}^7 \beta_i x_i \quad (2),$$

where y is the MMT, α is the intercept. x_1, \dots, x_i ($i=1, \dots, 7$) are the independent variables in Table 2, while β_1, \dots, β_i ($i=1, \dots, 7$) are the regression coefficients.

No.	Independent Variables
x_1	MFT (°C)
x_2	Annual mean temperature(°C)
x_3	78th percentile temperature (°C)
x_4	Latitude
x_5	GDP/capita (Dollars)
x_6	Annual temperature range (°C)
x_7	Study year (year)
Table 2 Independent Variables considered in the statistical analysis.	
Note: Study year is the mean year of study period.	

We used MLR to investigate the associations between MMTs and seven independent variables (Table 2). MLR results for different models are shown in Table 3. Since MFT and annual mean temperature, 78th percentile temperature, latitude was highly correlated, with correlation coefficients 0.72, 0.77 and -0.53 respectively (p-values<0.001). We have to only put one of them in the model to avoid the co-linearity.

Name	Model formula	Adjusted R ²	AIC
Model 1	$y = x_1$	0.84	1536
Model 2	$y = x_2$	0.51	2005
Model 3	$y = x_3$	0.60	1906
Model 4	$y = x_4$	0.28	2168
Model 5	$y = x_5$	0.19	2220
Model 6	$y = x_1 + x_5 + x_7 + x_8$	0.84	1539
Model 7	$y = x_2 + x_5 + x_7 + x_8$	0.61	1919
Model 8	$y = x_3 + x_5 + x_7 + x_8$	0.62	1897
Model 9	$y = x_4 + x_5 + x_7 + x_8$	0.32	2152
Table 3. A comparison of the results of multiple linear regression.			

By comparing the results of the different models (Table 3), Substantially, Model 1 displays the lowest AIC and highest R². The adjusted R² of Model 1 was 0.84, which was obviously higher than the adjusted R² of Model 2 (0.51), 3 (0.60) and 4 (0.28). Comparing with Model 1, R² and AIC of Model 6 changes a little with additional covariates (x_5 , x_7 and x_8). The coefficients of the model 6 are shown in Table S2. From Table S2, we can see that the association between MFT and MMT is not modified by socioeconomic conditions (GDP/capita) and the study year.

Parameters	Estimate	Std. Error	P-Value
MFT	0.84	0.02	< 2e-16***
Annual temperature range	0.01	0.01	0.295
GDP/capita	0.00	0.00	0.294
Study year	-0.007	0.02	0.657

Supplementary Table S2: Coefficients for the Model 6 based on 420 MMTs.

Comment B3:

3. On temporal variation of MMT and predictability of this variation in future. The authors claimed that they “can estimate the MMT in any geographical location [in any period.]” (Line 24-25)

However, to predict in time dimension is not identical to predict in space dimension. The manuscript showed only spatial variation of MMT estimated from empirical data (unfortunately, I could not find what statistical methods they used to estimate MMT in the manuscript). The manuscript did not include [empirical evidence on temporal variation of MMT in the past]. Of course, I agree that MMT may vary globally over time, but I could not find the evidence in References #1-8 either (cited in the paragraph starting with “However, MMT varies considerably across regions”. Whether MMT can vary over time must be empirically demonstrated first. Without this demonstration, how can the authors predict MMT in future?

Response B3: We have added relevant evidence in the discussion, as following (See lines 210-224):

In addition, we have added relevant evidence in the discussion, as following: “Several previous studies have shown that MMTs changed over time¹⁻⁴. For example, Donaldson found that MMTs rose consistently with the increase of annual mean temperature from 1971 to 1997, by 3.6°C in North Carolina and by 2.7°C in Southeast England and by 1.3°C in South Finland¹. Ekamper found that during the past 150 years, the MMT increased from slightly below 15°C to around 17°C in the Netherlands². Similar results have also been reported in other countries such as the United States and Europe. By analyzing the temperature-death relationships in 52 Spanish cities, Tobías found that MMTs rose almost exactly at the same rate as annual mean temperature (1°C/°C). This suggests that Spanish had broadly adapted to their local climate to the same extent⁴.”

“Acknowledging the matching between MMT and climate change, we estimate the global distribution of MMTs in the future. We found that, under RCP4.5 and RCP8.5 scenarios, compared to the 2010s, in the 2050s the global average MMT increase by 1.1°C (95%CI=0-2.5°C) and 1.8°C (95%CI=0.3-3.2°C) respectively (Figure 4).”

On the other hand, the study year in the present studies shows a wide range, from 1984 to 2014 years. By comparing the results of the Models 1 and 6 (Table

3), we can see that the association between MFT and MMT is not modified by the study period. So based on the association between MFT and MMT in present, it is reasonable to assume that the future MFT can represent the future MMT. However, we have added discussed the limitation in the revision as “Whether the future MFT can represent the future MMT is dependent upon the speed with which MMT may track changing MFT. There are still uncertainties for the speed of human adaptation of temperature in the future.” (See lines 261-264,).

Comment B4:

Here is an another issue. As suggested by the result of subtropical region, MFT is not as a strong predictor of MMT as is in the other regions. The authors stated “In subtropical region, matching is not as good as in the other two regions, partly due to the peak value of temperature distribution in some subtropical regions are not obvious. When the temperature distribution is relatively even throughout the year, people's adaptation to temperature is not very obvious.”(without any citation) (Line 171-174). To me, this result indicates very important limitations to predict MMT in the future. In future, temperature distribution may also change over time and climate zones may shift over time. Is it reasonable to assume that MFT can represent MMT in future, without investigating future change of temperature distributions in regions and climate zones? I think no.

Response B4: We thank the reviewer very much for the very good comments. We agree and have performed more analyses and improved the statement. We examined the distribution of projected daily mean temperature under RCP8.5 scenario in the 2050s compared to the 2010s in 15 representative cities (Figure R1). These 15 cities covered three major climatic zones (tropical, subtropical and temperate zones). Figure R1 shows that the temperature distribution of these cities changed very slightly from 2010s to 2050s.

Figure. R1 | The distribution of projected daily mean temperature under RCP8.5

scenario in the 2050s (red) compared to the 2010s (green) in 15 representative cities.

In the subtropical region, even though the correlations between MMT and MFT were smaller than those in tropical and temperate zones, R^2 still reaches 63% (and better than other two mainstream indicators (Figure 3)). Thus, we believe MFT is a simple and good predictor for MMT in all regions, particularly, in the scenario that future local temperature distribution would not change too much.

Moreover, we found that there is no systematic bias for MFT representing MMT in all climate zones (Figure 3). Therefore, even if temperature distribution may change or climate zones may shift over time, it is still reasonable to assume that future MFTs can represent future MMTs.

Reviewer #3 (Remarks to the Author):

This is an original and thoughtful contribution to the field of heat-health impact assessment in the context of a changing climate.

Response: We thank the reviewer very much for the encouraging comments. We have revised the paper according to your comments and suggestions as following.

Comment C1:

One major issue that is not addressed and which is central to the interpretation of this paper, is that of how the minimum mortality temperature (MMT) was computed in the studies that are included in their quantitative analysis. The authors seem to view the MMT as some sort of fundamental quantity of nature, when in fact it is just a summary metric that emerges from a particular statistical model. If the model were to change, the MMT might change. Thus it's important to add a section defining MMT as used here and describing how it was estimated in the included papers. Do the author's conclusions only apply to one particular MMT computation method? i.e., Were papers only included that used one method for deriving the MMT? That might explain why other important papers that have reported "MMTs" seem to be missing. Todd and Valeron EHP 2015 is one prominent example. Also, many of the China heat-mortality published by Tiantian Li and colleagues, e.g., Li et al., Scientific Reports 2016.

Response C1: We have supplemented the Method and References. See lines 64-72. "Time-series regression or time-stratified case-crossover models combining with distributed lag non-linear model (DLNM)⁶⁻⁹ are currently deemed as appropriate approaches to estimate temperature effects, and can provide comparable results. To make the results comparable, we only investigate 420 empirical MMTs which were modeled by DLNM."

We acknowledge that the different parameter specifications could cause uncertainties. Many studies have conducted sensitivity analyses to check the robustness of their findings. The analyses indicated that for the different parameter specifications, the results were slight different. We have added the limitation in the discussion (See lines 253-259).

The MMT was defined as the specific temperature associated to the lowest mortality risk. It is derived by the lowest point of the overall cumulative exposure-response curve (DLNM).

Minor comments:

Comment C2:

Line 22: Adaption is not a word.

Response C2: Sorry for the typo. We have changed to “adaptation”.

Comment C3:

Line 51: remove ? after MMT

Response C3: Done. Thanks. See line 59.

Comment C4:

Line 56: What were the exclusion criteria? Many papers that "investigated the relationships between temperature and human mortality counts for non-external causes" seem to be missing. So there must have been other exclusion criteria (e.g., "had to report MMT using method X").

Response C5: Done. See line 71.

"We reviewed a total of 210 papers, excluding the literatures from the repeated study areas. 16 papers of which investigated the relationships between temperature and human mortality counts for non-external causes using DLNM method."

Comment C5:

Line 136: replace "identical to" with "well approximated by"

Response C5: Done. See line 18.

Comment C6:

Line 142: delete "to our knowledge"

Response C6: Done. See Line 191.

Comment C7:

Line 162: include other citations that have questioned this, including:

Kinney et al., Environmental Research Letters 2015; Ebi and Mills, Climatic Change 2013; Staddon et al., Nature Climate Change 2014.

Response C7: We have added the recommended references¹⁰⁻¹² in the revision.

References

1. Donaldson, G.C., et al. Changes in summer temperature and heat-related mortality since 1971 in North Carolina, South Finland, and Southeast England. *Environmental Research*. **91**, 1–7 (2003).
2. Ekamper, P., et al. 150 Years of temperature-related excess mortality in the Netherlands. *Demographic Research*. **25**, 385–426 (2009).
3. Petkova, E.P., et al. Heat and mortality in New York City since the beginning of the 20th century. *Epidemiology*. **25**, 554–560 (2014).
4. Tobías, A., et al. Investigating Uncertainty in the Minimum Mortality Temperature. *Epidemiology*. **28**, 72–76 (2017).
5. McMichael T. Human Frontiers, Environments and Disease: Past patterns, uncertain Futures. *Cambridge University Press*. (2001).
6. Gasparrini, A., et al. Mortality risk attributable to high and low ambient temperature: a multicountry observational study. *Lancet* **386**, 369–375 (2015).
7. Yang, J., et al. Cardiovascular mortality risk attributable to ambient temperature in China. *Heart* **101**, 1966–1972 (2015)
8. Todd, N., and Valleron, A.J. Space–Time Covariation of Mortality with Temperature: A Systematic Study of Deaths in France, 1968–2009. *Environ Health Perspect* **123**, 659–664 (2015).
9. Li, T.T., et al. Heat-related mortality projections for cardiovascular and respiratory disease under the changing climate in Beijing, China. *Sci Rep* DOI: 10.1038/srep11441 (2015).
10. Kinney, P.L., et al. Winter season mortality: will climate warming bring benefits? *Environ. Res. Lett.* **10**, 111002 (2015).
11. Ebi, K. L., and Mills, D. Winter mortality in a warming climate: a reassessment. *WIREs Clim Change* **4**, 203–212 (2013).
12. Staddon, P.L., et al. Climate warming will not decrease winter mortality. *Nature Clim. Change* **4**, 190–194 (2014).

Reviewers' comments:

Reviewer #1 (Remarks to the Author):

This version of the ms is much improved, but I still feel the paper is trying to prove a thesis rather than dispassionately investigate it. Good science makes a proposition and explains its value, but then tries to build a case against it, to test its resilience.

Abstract: "Because MFTs will increase in the future, we predicted MMT will increase accordingly. Based on this finding, we..."

A prediction is not a "finding". You could instead say something like "We proceed on the assumption that this prediction is correct, and ..."

"MFT is the longest period of ambient temperature histogram which human expose to and therefore adapt to. So it is a very good indicator to reflect MMT." (repeated in lines 193-194) What do you mean by "longest period"? Please express this more carefully. For example, what kind of histogram (daily data? whole degrees Celsius? any smoothing?). If there is not enough space in the abstract for this level of detail, then I suggest delete from here. Also, the second sentence, "So ...", absolutely does not follow.

"The association between MFT and MMT was not modified by 20 socioeconomic conditions and period." What does 'period' mean here? The abstract should make sense on its own.

"Changeless" unchanging.

204-204 "Similar results have also been reported in other countries such as the United States and Europe." Refs please.

200- You cite several studies that support your thesis that MMT=MFT. Were there none with contrary or inconclusive findings? Please report a comprehensive survey of all previous reports on this question, which after all is critical to your paper. If you looked hard and found none that disagreed, then reporting that will greatly strengthen your case.

223 "we compared the associations between the MFT, the 78th quartile temperature" I suppose you mean the 78th percentile, not quartile (as in line 264 but not 279)- but why this strange choice?

286- "We used stepwise MLR to investigate the effects of each independent variable on the MMTs. Model selection was based on the lowest Akaike Information" In my opinion, with so few covariates (here just 7) there is no need or justification for using stepwise regression: better to just fit the full model and report those results. Little is gained by dropping non-significant variables, and something is lost because doing so can bias the estimation for those kept.

You defend MLR on the basis of collinearity (131-136). However, the correlations you report are not extremely high (0.77 and less), and with a reasonably large dataset, which you have, there should be no numerical instability in fitting the full model. Surely this would result in better estimation of MMT in the future, adjusting for mean temperature, latitude and so on. You may also wish to present a simplified model for practical use, but please for scientific purposes report the full model.

What happened to x6, temperature range? It is listed in Table 2, but appears in none of the models listed and is not mentioned again except in Table S2, which contradicts Table 2 (it's probably just a typo).

Reviewer #2 (Remarks to the Author):

I am very satisfied with Response B4.

1. Regarding temporal variation of MMT and of temperature

The authors stated "we can see that the association between MFT and MMT is not modified by the study period. So based on the association between MFT and MMT in present, it is reasonable to assume that the future MFT can represent the future MMT" in Response B3. I am curious about what the study period variable exactly means? Is this an indicator of the study period for each location to estimate MMT?. If it is, I think the result of the study year variable may not support the argument. To me, this result just represents that the association of spatial variation of MMT and of MFT is not confounded by the study year.

I hazard a guess that if the study year variable well represents a variation of MMT (if exists) at different locations tied with different study periods, the result would have indicated that a temporal variation of MMT may exist in almost 400 locations (this seems an indirect testing for a temporal variation of MMT because this is not a comparison of MMTs of different study periods at the same location.) If the condition is correct, Table 2 would not support the existence of a temporal variation of MMT.

Reviewer #3 (Remarks to the Author):

The responses to my comments have improved the quality of the manuscript. I am satisfied with the result.

Reviewers' comments:

Reviewer #1 (Remarks to the Author):

This version of the ms is much improved, but I still feel the paper is trying to prove a thesis rather than dispassionately investigate it. Good science makes a proposition and explains its value, but then tries to build a case against it, to test its resilience.

Response: We thank the reviewer very much for this suggestion. We have improved the paper according to your suggestion.

Comment A1:

Abstract: “Because MFTs will increase in the future, we predicted MMT will increase accordingly. Based on this finding, we…”

A prediction is not a “finding”. You could instead say something like “We proceed on the assumption that this prediction is correct, and …”

Response A1: We have changed the wording of the statement to: ‘Based on the MFT~MMT association, we estimated and mapped the global distribution of MMTs in the present (2010s) and the future (2050s) for the first time. We found that MMT will increase throughout the world in the future.’ (See lines 24-27 and 191-194).

We also have carefully checked and improved the presentation throughout the paper.

Comment A2:

“MFT is the longest period of ambient temperature histogram which human expose to and therefore adapt to. So it is a very good indicator to reflect MMT.” (repeated in lines 193-194)

What do you mean by “longest period”? Please express this more carefully. For example, what kind of histogram (daily data? whole degrees Celsius? any

smoothing?). If there is not enough space in the abstract for this level of detail, then I suggest delete from here. Also, the second sentence, "So ...", absolutely does not follow.

Response A2: We mean that: In the daily mean temperature histogram for a given year, MFT is the most frequently exposed temperature for people. MFT is estimated by the whole degrees Celsius.

We have deleted this information from the abstract because of limited space.

Comment A3:

“The association between MFT and MMT was not modified by socioeconomic conditions and period.” What does ‘period’ mean here? The abstract should make sense on its own.

Response A3: “period” is the mean year of the study period. For example, a study using data between 2000 and 2010 would have a study year of 2005.

We have improved the statement as “The association between MFT and MMT was not modified by socioeconomic conditions and study year” (See lines 22-24).

Comment A4:

“Changeless” unchanging.

Response A4: Done. Thank you. See line 33.

Comment A5:

204-204 “Similar results have also been reported in other countries such as the United States and Europe.” Refs please.

Response A5: We have added references in the revision, as ‘Similar results have also been reported in other countries such as the United States³⁰ and Europe³¹.’ See line 258.

Comment A6:

200- You cite several studies that support your thesis that MMT=MFT. Were there none with contrary or inconclusive findings? Please report a comprehensive survey of all previous reports on this question, which after all is critical to your paper. If you looked hard and found none that disagreed, then reporting that will greatly strengthen your case.

Response A6: We searched Web of Science, PubMed, Scopus and EMBASE for

articles published in English from January 1990 to May 2019 using the terms ‘temperature’ or ‘heat’, and ‘death’ or ‘mortality’, and “temporal change” or “temporal variation” or “over time”. A total of 13 papers¹⁻¹³ investigated the changes of relationships between daily temperature and human mortality counts for non-external causes over time. All papers directly or indirectly indicate that MMT changed over time. This finding could strength our conclusions.

Comment A7:

223 “we compared the associations between the MFT, the 78th quartile temperature” I suppose you mean the 78th percentile, not quartile (as in line 264 but not 279) – but why this strange choice?

Response A7: We are sorry for the typo. We have replaced “quartile” by “percentile” (See Line 155). We have carefully checked throughout the paper.

Several studies indicated that MMT fluctuates around a certain percentile of the local temperature (minimum mortality percentile, MMP) ^{14,15}. The mean MMP of the 420 locations was 78th. Therefore, we chose the 78th percentile of temperature, the annual mean temperature and MFT to compare their associations with MMT.

Comment A8:

286- “We used stepwise MLR to investigate the effects of each independent variable on the MMTs. Model selection was based on the lowest Akaike Information” In my opinion, with so few covariates (here just 7) there is no need or justification for using stepwise regression: better to just fit the full model and report those results. Little is gained by dropping non-significant variables, and something is lost because doing so can bias the estimation for those kept.

You defend MLR on the basis of co-linearity (131-136). However, the correlations you report are not extremely high (0.77 and less), and with a reasonably large dataset, which you have, there should be no numerical instability in fitting the full model. Surely this would result in better estimation of MMT in the future, adjusting for mean temperature, latitude and so on. You may also wish to present a simplified model for practical use, but please for scientific purposes report the full model.

Response A8: We thank the reviewer very much for the comments. We have deleted the co-linearity consideration and changed the method to a multiple linear regression (MLR) model. We used an MLR to investigate and compare the fitting effect of different models. The MLR results are shown in Table 2. Model 1 is a full model. Models 2–4 only consider one independent variable.

Name	Model formula	Adjusted R ²	AIC
Model 1	MMT ~ MFT + Annual mean temperature + 78th percentile temperature + Latitude + GDP/capita + Annual temperature range + Study year	0.86	1501
Model 2	MMT ~ MFT	0.84	1531
Model 2	MMT ~ Annual mean temperature	0.51	2005
Model 3	MMT ~ 78th percentile temperature	0.60	1906
Model 4	MMT ~ Latitude	0.28	2168
Table 2. The results of multiple linear regressions.			

From the above results, we can see that Model 1 displays the lowest AIC and highest R². The adjusted R² of Model 1 was 0.86. In addition to this, we found that if we only used MFT as predictor (Model 2), the predictive ability is as good as model 1 (the adjusted R²: 0.84). Model 2 performed much better than that of using Annual mean temperature (Model 3, adjusted R² = 0.51), 78th percentile temperature (Model 4, adjusted R² = 0.60), and latitude (Model 5, adjusted R² = 0.28), respectively. However, to present a simplified model for practical use, we recommended using Model 2 to predict MMT. See lines 105-120.

Comment A9:

What happened to x6, temperature range? It is listed in Table 2, but appears in none of the models listed and is not mentioned again except in Table S2, which contradicts Table 2 (it's probably just a typo).

Response A9: We are sorry for the typo. We have revised the subscript (Table 2). See line 104.

Reviewer #2 (Remarks to the Author):

I am very satisfied with Response B4.

Comment B1:

1. Regarding temporal variation of MMT and of temperature

The authors stated “we can see that the association between MFT and MMT is not modified by the study period. So based on the association between MFT and MMT in present, it is reasonable to assume that the future MFT can represent the future MMT” in Response B3. I am curious about what the study period variable exactly means? Is

this an indicator of the study period for each location to estimate MMT? If it is, I think the result of the study year variable may not support the argument. To me, this result just represents that the association of spatial variation of MMT and of MFT is not confounded by the study year.

I hazard a guess that if the study year variable well represents a variation of MMT (if exists) at different locations tied with different study periods, the result would have indicated that a temporal variation of MMT may exist in almost 400 locations (this seems an indirect testing for a temporal variation of MMT because this is not a comparison of MMTs of different study periods at the same location.) If the condition is correct, Table 2 would not support the existence of a temporal variation of MMT.

Response B1: The reviewer is right. The “period” is the mean year of study period. For example, a study using data between 2000 and 2010 would have a study year of 2005. This method is used to check that the association between MFT and MMT is not confounded by the study year, and that only using MFT is good enough to predict MMT.

To check if MMTs have changed over time, we have to get at least two MMTs at different years from the same location. However, only one paper¹ provides the MMT values in different years at three areas (North Carolina, Southeast England and South Finland). We calculated the MFTs of these three locations in the corresponding years, and compared them with MMTs. The data on daily mean temperature of these three locations during the study period was downloaded from the United States National Oceanic and Atmospheric Administration’s (NOAA) website (<https://www.ncdc.noaa.gov/>). We found that the changing trend of MFT is the same as MMT (Supplementary Fig. S3). We have added these results in the discussion, to support our conclusion. See lines 254-257.

(a) North Carolina

(b) Southeast England

(c) South Finland

Supplementary Fig. S3 | Comparison of changes in MMTs and MFTs in three locations from 1971 to 1997.

Reviewer #3 (Remarks to the Author):

The responses to my comments have improved the quality of the manuscript. I am satisfied with the result.

Response: We thank the reviewer very much for the encouraging comments.

References

1. Donaldson, G. C. *et al.* Changes in summer temperature and heat-related mortality since 1971 in North Carolina, South Finland, and Southeast England. *Environ. Research* **91**, 1–7 (2003).
2. Ekamper, P. *et al.* 150 Years of temperature-related excess mortality in the

- Netherlands. *Demographic. Research* **25**, 385–426 (2009).
3. Petkova, E. P. *et al.* Heat and mortality in New York City since the beginning of the 20th century. *Epidemiology* **25**, 554–560 (2014).
 4. Ha, J. & Kim, H. Changes in the association between summer temperature and mortality in Seoul, South Korea. *Int. J. Biometeorol* **57**: 535-544 (2013).
 5. Matzarakis, A. *et al.* Human biometeorological evaluation of heat-related mortality in Vienna. *Theor. Appl. Climatol* **105**:1–10 (2011).
 6. Donaldson, G.C. & Keatinge, W.R. Decline in cold-related ischaemic heart, cerebrovascular, respiratory disease, and all-cause mortalities, between 1979 and 1994 in south-east England. *Br. Med. J* **315**: 1055 - 1056 (1997).
 7. Smoyer, K.E. *et al.* A comparative analysis of heat waves and associated mortality in St Louis, Missouri—1980 and 1995. *Int. J. Biometeorol* **42**: 44 - 50. (1998).
 8. Wannamethee, S. G. *et al.* Declining vulnerability to temperature-related mortality in London over the 20th century. *Diabetologia* **53**:890–898 (2010).
 9. Lerchl, A. Changes in the seasonality of mortality in Germany from 1946 to 1995: the role of temperature. *Int. J. Biometeorol* doi:10.1007/s004840050089 (1998).
 10. G Marcuzzi, M Tasso. Seasonality of death in the period 1889-1988 in the Val di Scalve. *Human Biology* doi: 10.1016/0888-7543(92)90329-Q (1992).
 11. Barnett & Gerard, A. Temperature and cardiovascular deaths in the US elderly: changes over time. *Epidemiology* doi: 10.1097/01.ede.0000257515.34445.a0 (2007).
 12. Davis, R. E. Changing heat-related mortality in the United States. *Environ. Health. Perspect* **111**:1712–1718 (2003).
 13. Linnarsjö, A. *et al.* Acute fatal effects of short-lasting extreme temperatures in Stockholm, Sweden: evidence across a century of change. *Int. J. Cardiol* doi: 10.1097/01.ede.0000434530.62353.0b (2000).
 14. Yang, J. *et al.* Cardiovascular mortality risk attributable to ambient temperature in China. *Heart* **101**, 1966–1972 (2015).
 15. Guo, Y. M. *et al.* Global variation in the effects of ambient temperature on mortality: a systematic evaluation. *Epidemiology* **25**, 781–789 (2014).

Response to reviewers

Reviewers' comments:

Reviewer #2 (Remarks to the Author):

I have found the manuscript has improved. I much appreciate it.

Response: Thanks very much for the encouraging comments.

Comment B1.

The authors stated in the Abstract as follows:

“The association between MFT and MMT was not significantly modified by socioeconomic conditions, annual mean temperature, latitude, and study year. “

(I also find that the same sentence in the body of the manuscript”

The expression “modify the association between X and Y” is confusing.

This expression sounds about effect modification or interaction, which the authors did not test.

As far as I understand the previous manuscripts and the authors’ response to the reviewers, the authors showed that the association between MMT and factors other than MFT was negligible and the association between MMT and MFT was not changed much.

If the authors want to stress out that the association between MMT and MFT is stable even after adjusting for socioeconomic conditions and the other factors, I would suggest other expressions like “the association between MMT and MFT was not changed when [factors] were adjusted”

Also, I am worried whether “study year” is okay to come out in the Abstract. Readers would be confused about the meaning of “study year”.

Response B1: (1) We have improved the statement as “The association between MFT and MMT is not changed when we adjust for socioeconomic conditions and latitude.” See lines 26-28 and 195-196.

(2) We have deleted ‘study year’ from the abstract according to the reviewer’s suggestion.

Comment B2.

In Line 206-224, the authors discussed previous studies which have reported only temporal increases of MMT. There are also conflicting findings against their

hypothesis. Gasparrini et al 2015 have found that MMT increased in some countries but did not in other countries (e.g. United Kingdom and South Korea). Temporal variation of MMT differed across cities in Spain (Miron et al 2008) and in South Korea (Kim et al 2019): MMTs in some cities increased but others decreased or not changed. The authors should discuss conflicting findings and whether that temporal variation of MMT may increase in every places or not.

Gasparrini, A., Guo, Y., Hashizume, M., Kinney, P. L., Petkova, E. P., Lavigne, E., ... & Tong, S. (2015). Temporal variation in heat – mortality associations: a multicountry study. *Environmental health perspectives*, 123(11), 1200-1207.

Miron, I. J., Criado-Alvarez, J. J., Diaz, J., Linares, C., Mayoral, S., & Montero, J. C. (2008). Time trends in minimum mortality temperatures in Castile-La Mancha (Central Spain): 1975 – 2003. *International journal of biometeorology*, 52(4), 291-299.

Kim, H., Kim, H., Byun, G., Choi, Y., Song, H., & Lee, J. T. (2019). Difference in temporal variation of temperature-related mortality risk in seven major South Korean cities spanning 1998 – 2013. *Science of the Total Environment*, 656, 986-996.

Response B2: We thank the reviewer for the comments. We are sorry that we did not make a precise expression in the last version.

(1) The reviewer is right. The existing several studies have found that temporal variation of MMT differed across regions, MMTs in some cities increased but others decreased or not changed over time. This phenomenon is similar to the annual average temperature. In the context of global warming, although most land areas have experienced warming, but there are still a few areas where temperature varies little or even decreases (Wang, et al., 2018), as well as MFT. By comparing the global MFT changes between 2010s to 2050s, we found that under the two scenarios (RCP4.5 and RCP8.5), most of the land regions will experience increased MFT, but there are a few areas where the temperature changes little or decreases. Under the two scenarios, the grids with elevated MFTs account for 94% and 96% of the total grids. Based on Figure 5, we calculate the percentage of the grid with different MFT variation ranges in the total global land area (Table R1).

	MFT variation range (°C)	Percentage (%)
MFTs decrease	[-4, -2]	0.2
	(-2, -1]	1.1
	(-1, 0]	4.7
MFTs increase	(0, 1]	42.1
	(1, 2]	40.3
	(2, 3]	9.5
	(3, 4.5]	2.1
	Total	[-4, 4.5]

Table R1. The percentage of the grid with different MFT variation ranges in the total global land area.

(2) On the other hand, the MMTs of some areas mentioned in the above literature changes are not obvious, another possible reason is that their study period is relatively short (1998-2013, 1993-2006, 1992-2010).

We have improved the statements in the revision. We have replaced “increase” by “change” (See Line 217).

1. Wang, J.F., Xu, C.D., Hu, M.G., et al. Global land surface air temperature dynamics since 1880. *Int. J. Climatol.* 38 (Suppl.1), e466 – e474. (2018).

Comment B3.

The authors stated in the Abstract as follows:

“We found that MMT will increase throughout the world in the future, in the context of global climate change.”

In Line 227, the authors also stated: “Third, in the context of global warming, MMT will increase in the future”.

I think that it is too strong to conclude “MMT will increase throughout the world”, as I pointed out in Comment 2. The authors also stated “However, whether the future MFT can represent the future MMT is dependent upon the speed with which MMT tracks the changing MFT. There are still uncertainties regarding the speed of human adaptation to temperature in the future.” in the discussion section. I would suggest that the authors should fine-tune their interpretations and conclusions.

Response B3: Because there is limited space to explain it in detail, we have deleted the sentence ‘We found that MMT will increase in the future, in the context of global climate change.....’ from the abstract.

In the discussion (line 237), we have improved the expressions as “Third, in the context of global warming, MMT may increase in the future.”

We also have carefully checked and improved the presentation throughout the paper.

Comment B4.

The authors stated in the Abstract as follows:

“Instead of using existing projections of temperature-related mortality based on unchanging MMT, future MMT predicted by MFT would provide more precise predictions.”

The authors stated that “more precise predictions.” What are predictions about? Health outcomes? Please clarify

Response B4: We mean prediction on mortality risks/burden due to ambient temperatures.

However, because there is word limit in abstract, we have deleted the sentence ‘Instead of using existing projections of temperature-related mortality based on unchanging MMT, future MMT predicted by MFT would provide more precise predictions’ from the abstract.

In addition, we have improved the statement as “Instead of using unchanging MMT, future MMT predicted by MFT might provide more precise predictions on the mortality risks/burden due to ambient temperatures.” in the discussion. See lines 240-242.

Reviewers' comments:

Reviewer #2 (Remarks to the Author):

-Comment 1

The first result of the manuscript is about the spatial association between MMT and MFT. The second result is about the prediction of MMT by using the association found as if it represents the temporal association between MMT and MFT, which was not tested in this study. For the second, the authors must assume that the spatial association can be transformed into (at least consistent with) the temporal association for the prediction. To advocate this assumption, the authors tried to discuss 1) whether MMT is varying over time 2) whether this temporal variation of MMT is associated with MFT, citing previous studies. I have pointed out problems of this assumption since the first review.

In the last review, I pointed out that although some findings have shown that MMT may vary over time, other findings have shown that it may not in some cities and countries (e.g., UK, Spain, South Korea). The authors replied that MMT may not vary over time in some cities as average temperature and MFT may not increase throughout all cities and countries. However, they did not reply that MFT (and average temperature) increased or not in the UK, Spain, South Korea. In addition, they replied that the latter findings may be due to limited study periods. But I find that the former findings are also partly based on studies with limited study periods as long as about 20-30 years.

The authors showed that MMT may be temporally associated with average temperature and MFT in North Carolina, South England, and South Finland and stated similar findings existed in other studies in Line 209-228 of the revised manuscript. However, when I checked citations which are added during the review process (Donaldson, G. C. et al.(2003), Ekamper, P. et al.(2009), Petkova, E. P. et al.(2014), Tobías, A. et al.(2017)), I could not find similar findings in the Netherland study Ekamper, P. et al.(2009) and the New York study Petkova, E. P. et al.(2014) as they wrote in the manuscript, which are studies with the two longest study period.

These two studies showed that MMT changed over time. But when I looked into changes in temperature distributions, it seems that the average temperature is not positively associated with the optimum temperature in the Netherlands for 150 years (MFT variation is not reported in this study). In the New York study, it can be drawn that MMT is associated with the average temperature. However, it seems due to increases in very high percentiles of temperature over time. It is not reported whether MFT was increased or not in New York. But I may imagine that MFT may not dramatically vary over time because the 90th temperature was very consistent over time (As far as I know, MMT is differently located around the 70-90th percentile over cities and countries). Average temperature is highly affected by extreme temperature value while an MFT is unlikely to be because it is a mode.

In summary, I feel that that the authors might have picked up, in the discussion section, only findings showing that 1) MMT varied over time and 2) MMT may be associated with MFT, without comprehensive discussions about the conflicting results and the conflicting explanations.

- Comment 2

While checking citations, I feel I have to point out again whether it is okay that discussions about socioeconomic conditions can be ruled out. Previous studies have explained temporally decreasing heat effect including increasing MMT as a population may adapt socioeconomically (Donaldson, G. C. et al.(2003), Ekamper, P. et al.(2009), Petkova, E. P. et al.(2014)). The authors emphasize that a population may be "acclimatized" to MFT, explained by Darwin's Theory, and focus on only "human biological evolution" (Line 203). According to the spatial association between MMT and MFT the authors found, this emphasis may be okay. However, the problem arises when the authors assume that the spatial association can be transformed into the temporal association to predict future MMT. The authors' prediction is based on the untested assumption. Previous studies

have recognized that socioeconomic changes are important for populations to adapt.

Donaldson, G. C. et al. Changes in summer temperature and heat-related mortality since 1971 in North Carolina, South Finland, and Southeast England. *Environ. Research* 91, 1–7 (2003).
Ekamper, P. et al. 150 Years of temperature-related excess mortality in the Netherlands. *Demographic. Research* 25, 385–426 (2009).
Petkova, E. P. et al. Heat and mortality in New York City since the beginning of the 20th century. *Epidemiology* 25, 554–560 (2014).
Tobías, A. et al. Investigating Uncertainty in the Minimum Mortality Temperature. *Epidemiology* 28, 72–76 (2017).

Reviewers' comments:

Reviewer #2 (Remarks to the Author):

Comment B1.

The first result of the manuscript is about the spatial association between MMT and MFT. The second result is about the prediction of MMT by using the association found as if it represents the temporal association between MMT and MFT, which was not tested in this study. For the second, the authors must assume that the spatial association can be transformed into (at least consistent with) the temporal association for the prediction. To advocate this assumption, the authors tried to discuss 1) whether MMT is varying over time 2) whether this temporal variation of MMT is associated with MFT, citing previous studies. I have pointed out problems of this assumption since the first review.

In the last review, I pointed out that although some findings have shown that MMT may vary over time, other findings have shown that it may not in some cities and countries (e.g., UK, Spain, South Korea). The authors replied that MMT may not vary over time in some cities as average temperature and MFT may not increase throughout all cities and countries. However, they did not reply that MFT (and average temperature) increased or not in the UK, Spain, South Korea. In addition, they replied that the latter findings may be due to limited study periods. But I find that the former findings are also partly based on studies with limited study periods as long as about 20-30 years.

The authors showed that MMT may be temporally associated with average temperature and MFT in North Carolina, South England, and South Finland and stated similar findings existed in other studies in Line 209-228 of the revised manuscript. However, when I checked citations which are added during the review process (Donaldson, G. C. et al.(2003), Ekamper, P. et al.(2009), Petkova, E. P. et al.(2014), Tobias, A. et al.(2017)), I could not find similar findings in the Netherland study Ekamper, P. et al.(2009) and the New York study Petkova, E. P. et al.(2014) as they wrote in the manuscript, which are studies with the two longest study period.

These two studies showed that MMT changed over time. But when I looked into changes in temperature distributions, it seems that the average temperature is not positively associated with the optimum temperature in the Netherlands for 150 years (MFT variation is not reported in this study). In the New York study, it can be drawn that MMT is associated with the average temperature. However, it seems due to increases in very high percentiles of temperature over time. It is not reported whether MFT was increased or not in New York. But I may imagine that MFT may not dramatically vary over time because the 90th temperature was very consistent over time (As far as I know, MMT is differently located around the 70-90th percentile over cities and countries). Average temperature is highly affected by extreme temperature value while an MFT is unlikely to be because it is a mode.

In summary, I feel that that the authors might have picked up, in the discussion

section, only findings showing that 1) MMT varied over time and 2) MMT may be associated with MFT, without comprehensive discussions about the conflicting results and the conflicting explanations.

Response B1: We thank the reviewer for the comments. We have to point out that we didn't only pick up the supporting evidence in the last version. We only cited some papers because of the word limitation. Here we would like to take this opportunity to discuss the evidence for the temporal change of MMT and MFT.

We have searched all available studies which provided the values of MMTs at different times in the same location. We found there were six papers (Please see details below), which provided the MMT values of 62 locations from eight countries in different years. We have calculated the MFTs of these locations during the same time, and have compared the temporal trends of MMT and MFT (please see below)..

1. Changes of MMT (left) and MFT (right) in **Japan, from 1972 to 2012** (Chung, Y. et al; 2018).

We can see that in Japan, from 1972 to 2012, the temporal trends of MMT and MFT are consistent.

2. Changes of MMT and MFT in **France, from 1968 to 2009** (Todd, N. et al., 2015).

Year	MMT (°C)	MFT (°C)
1968-1981	17.5	17.3
1982-1995	17.8	17.9
1996-2009	18.2	18.5

We can see that in France, from 1968 to 2009, the temporal trends of MMT and MFT are consistent.

3. Changes of MMT (left) and MFT (right) in **North Carolina (USA), from 1972 to 1997** (Donaldson, G. C. et al., 2003).

We can see that in North Carolina, from 1972 to 1997, the temporal trends of MMT and MFT are consistent.

4. Changes of MMT (left) and MFT (right) in **Southeast England, from 1972 to 1997** (Donaldson, G. C. et al., 2003).

We can see that in Southeast England, from 1972 to 1997, the temporal trends of MMT and MFT are consistent.

5. Changes of MMT (left) and MFT (right) in **South Finland, from 1972 to 1997** (Donaldson, G. C. et al., 2003).

We can see that in South Finland, from 1972 to 1997, the temporal trends of MMT and MFT are consistent.

and MFT are consistent.

6. Change of MMTs and MFTs in Spain, from 1980 to 2016 (Achebak, H. et al., 2019).

Figure: Change of MMTs in Spain from 1980-1994 to 2002-2016.

Figure: Distribution of daily mean temperatures in Spain between 1980 and 1994 and 2002 and 2016

From the above two figures (They are Figures 1 and 3 in Achebak, H. et al., 2019), we can see that in Spain, from 1980-94 to 2002-16, MMT rose from 19.5°C to 20.3°C, while MFT rose from 19.5°C to 20.5°C, the temporal trends of MMT and MFT are consistent.

7. Kim, H. et al studied the changes of MMT and MFT in 7 major South Korean, from 1998 to 2013 (Kim, H. et al., 2019). The changing-trend of MMT and MFT in 4 cities is consistent, while another 3 cities are not consistent.

8. Change of MMTs in Netherlands, from 1855 to 2006 (Ekamper, P. et al., 2009).

Year	MMT(°C)
------	---------

1855-1879	12.8
1880-1904	13
1905-1929	17.5
1930-1954	18.5
1955-1979	18
1980-2006	18

From the above results, we can see that the MMT of Netherlands has increased from 1855 to 2006. Since the daily temperature data of the Netherlands from 1855 to 1980 is not obtained, the MFT is not calculated.

9. Change of MFTs in New York, from 1901 to 2000.

The result shows that the MFT of New York has increased from 1901 to 2000. We did not find the existing study (the reviewer suggested) which provided the values of MMT in different years.

Results from all of the above countries show:

- 1) In the past few decades, the MMT and MFT have increased in most locations of the studied countries. However, there are also a few locations where MMT and MFT did not change or reduced.
- 2) In most locations, the temporal trends of MMT and MFT are consistent.

The reason for the inconsistent temporal trends of MMT and MFT in very few locations might be the random error. We cannot make a perfect case (MMT and MFT have the same trend in all locations) for the real world data. This is why researchers need to do data collection and perform statistical analysis.

In addition, when predicting the temperature-mortality relationships in the future, the existing studies have generally used an unchanging MMT as a reference value. This is impossible, as many studies have shown that MMT have changed, particularly increased in many locations. Our study provides a simple method to calculate future MMT by MFT, which provides a potential future scenario. This would provide

improved predictions on the mortality risks/burden due to ambient temperatures.

We acknowledge that our study still has limitation due to some uncertainties, although the existing research results show that in most of the studied locations MMT and MFT have the similar increasing-trend in the context of global climate change. However, this limitation exists in all studies on future projections, because we all don't know what will happen in the future. And our method provides a possible future scenario to improve the projection for future temperature-mortality associations.

We have added these evidences, references and limitation to the discussion section. See lines 211-235, 273-275 and 407-421.

Comment B2.

While checking citations, I feel I have to point out again whether it is okay that discussions about socioeconomic conditions can be ruled out. Previous studies have explained temporally decreasing heat effect including increasing MMT as a population may adapt socioeconomically (Donaldson, G. C. et al.(2003), Ekamper, P. et al.(2009), Petkova, E. P. et al.(2014)). The authors emphasize that a population may be “acclimatized” to MFT, explained by Darwin's Theory, and focus on only “human biological evolution” (Line 203). According to the spatial association between MMT and MFT the authors found, this emphasis may be okay. However, the problem arises when the authors assume that the spatial association can be transformed into the temporal association to predict future MMT. The authors' prediction is based on the untested assumption. Previous studies have recognized that socioeconomic changes are important for populations to adapt.

Response B2: We are sorry that we did not make a precise expression in the last version.

We believe that socioeconomic conditions affect human adaptation to temperature. Specifically, socioeconomic conditions indeed affect the mortality risks (relative risks) associated with temperature. For example, people with low socioeconomic levels are more vulnerable and have a higher mortality risk associated with temperature. However, whether the MMT is influenced by socioeconomic conditions is still not clear. In our analysis, we found that socioeconomic condition (GDP/capita) was not associated with MMT. And the association between MFT and MMT is not changed when we adjust for socioeconomic conditions, which means that MFT-MMT association is independent to the socioeconomic conditions. For example, as shown in the figures below, in high GDP regions, such as Canada (left, 20 cities), MMT can be well represented by the local MFT. While, in regions with low GDP, such as Thailand (right, 62 cities), MMT can still be well represented by the local MFT.

In order to validate this conclusion, we applied a standard multiple linear regression model (MLR) to fit the MMT according to the seven independent variables (Table 1). Model is defined as follows (Eq.(2)):

$$y = \alpha + \sum_{i=1}^7 \beta_i x_i \quad (2),$$

where y is the MMT, α is the intercept. x_1, \dots, x_7 ($i=1, \dots, 7$) are the independent variables in Table 2, while β_1, \dots, β_7 ($i=1, \dots, 7$) are the regression coefficients.

No.	Independent variables
x_1	MFT (°C)
x_2	Annual mean temperature (°C)
x_3	78 th percentile temperature (°C)
x_4	Latitude (°)
x_5	GDP/capita (USD)
x_6	Annual temperature range (°C)
x_7	Study year (year)

Table 1 Independent variables considered in the statistical analysis.

Note: Study year is the mean year of study period.

MLR results for different models are shown in Table 2.

Name	Model formula	Adjusted R ²	AIC
Model 1	MMT ~ MFT + Annual mean temperature + 78th percentile temperature + Latitude +GDP/capita + Annual temperature range + Study year	0.86	1501
Model 2	MMT ~ MFT	0.84	1531
Model 3	MMT ~ Annual mean temperature	0.51	2005
Model 4	MMT ~ 78th percentile temperature	0.60	1906
Model 5	MMT ~ Latitude	0.28	2168

Table 2. The results of multiple linear regressions.

AIC: Akaike Information Criterion.

From the above results, we can see that Model 1 displays the lowest AIC and highest R². The adjusted R² of Model 1 was 0.86. In addition to this, we found that if we only

used MFT as predictor (Model 2), the prediction is almost as good as Model 1 (the adjusted R^2 : 0.84) but much simpler than the latter.

The coefficients of Model 1 are shown in Supplementary Table 1. From Supplementary Table 1, we can see that the association between MFT and MMT is not changed when socioeconomic conditions (gross domestic product [GDP]/capita), is adjusted. Therefore, to present a simplified model for practical use, we recommend using Model 2 to predict MMT.

Parameters	Estimate	Std. error	P-value
MFT	0.70	0.03	< 2e-16 ***
Annual mean temperature	0.10	0.03	0.013*
78th percentile temperature	0.09	0.03	0.017*
Latitude	0.01	0.01	0.170
Annual temperature range	0.05	0.01	0.295
GDP/capita	0.00	0.00	0.379
Study year	0.05	0.02	0.299
Supplementary Table 1: Coefficients for Model 1 based on 420 MMTs.			

References:

- [1]. Chung, Y. et al. Changing susceptibility to Non-Optimum Temperatures in Japan, 1972–2012: The Role of Climate, Demographic, and Socioeconomic Factors. *Environ. Health. Perspect* doi: 10.1289/EHP2546 (2018).
- [2]. Donaldson, G. C. et al. Changes in summer temperature and heat-related mortality since 1971 in North Carolina, South Finland, and Southeast England. *Environ. Res* **91**, 1 – 7 (2003).
- [3]. Todd, N & Valleron, A. J. Space–Time Covariation of Mortality with Temperature: A Systematic Study of Deaths in France, 1968–2009. *Environ. Health. Perspect* **123**, 659-664 (2015).
- [4]. Achebak, H. et al. Trends in temperature-related age-specific and sex-specific mortality from cardiovascular diseases in Spain: a national time-series analysis. *Lancet Planet Health* doi: 10.1016/S2542-5196(19)30090-7 (2019).
- [5]. Kim, H. et al. Difference in temporal variation of temperature-related mortality risk in seven major South Korean cities spanning 1998–2013. *Sci Total Environ* **656**, 986 – 996 (2019).
- [6]. Ekamper, P. et al. 150 Years of temperature-related excess mortality in the Netherlands. *Demographic. Research* **25**, 385–426 (2009).

Reviewers' comments:

Reviewer #2 (Remarks to the Author):

1. I am satisfied with Response B1. I appreciate that the authors have improved the paragraph (Line 209-232).

2. Although I understood that the authors may say that socioeconomic conditions were not associated with MMT, their results are not saying that socioeconomic [change (over time)] is not associated with MMT.

The authors did not test whether socioeconomic [changes over time] was associated with a temporal change of MMT.*

But I agree that MFT may be a good variable to predict MMT.

*I understand that the authors may not be able to test this for a long period of time for a global scale.

Since this paper may echo many readers, I think that mechanisms of population adaptation to temperature should be carefully described. By reading Line 181-207, messages may be taken such as "human would be (physiologically) acclimatized to temperature". This is unlikely to be the only pathway for adaptation to high temperature. (McMichael et al 2006) I would suggest that instead of emphasizing results about a GDP variable, the authors should briefly introduce (cite) other mechanisms in addition to physiological acclimatization.

McMichael, A. J., Woodruff, R. E., & Hales, S. (2006). Climate change and human health: present and future risks. *The Lancet*, 367(9513), 859-869.

Reviewers' comments:

Reviewer #2 (Remarks to the Author):

Comment B1.

I am satisfied with Response B1. I appreciate that the authors have improved the paragraph (Line 209-232).

Response B1: Thanks.

Comment B2.

Although I understood that the authors may say that socioeconomic conditions were not associated with MMT, their results are not saying that socioeconomic [change (over time)] is not associated with MMT.

The authors did not test whether socioeconomic [changes over time] was associated with a temporal change of MMT.*

But I agree that MFT may be a good variable to predict MMT.

*I understand that the authors may not be able to test this for a long period of time for a global scale.

Since this paper may echo many readers, I think that mechanisms of population adaptation to temperature should be carefully described. By reading Line 181-207, messages may be taken such as "human would be (physiologically) acclimatized to temperature". This is unlikely to be the only pathway for adaptation to high temperature. (McMichael et al 2006) I would suggest that instead of emphasizing results about a GDP variable, the authors should briefly introduce (cite) other mechanisms in addition to physiological acclimatization.

McMichael, A. J., Woodruff, R. E., & Hales, S. (2006). Climate change and human health: present and future risks. *The Lancet*, 367(9513), 859-869.

Response B2: Thanks for the great suggestions. Indeed, humans adapt to climate in several ways, such as physiological, behavioral, and technological adaptations (McMichael, 2006; Guo, 2014; Guo, 2018). This study of MFT may reveal an important aspect of how humans physiologically adapt to ambient temperature.

In addition to physiological acclimatization, we believe that socioeconomic conditions can affect human's behavioral and technological adaptations to ambient temperature. For example, people with low socioeconomic levels are more vulnerable and have a higher mortality risk associated with temperature (Scovronick, 2018; Chen, 2015). However, whether the MMT is influenced by socioeconomic conditions [change (over time)] is still not clear. And it is hard to test because of the long period of data is not easy to obtain. We only find one paper which studied the effects of changing socioeconomic conditions on temperature-mortality association, including residential

air conditioning, electrification (represented by residential electrification) and access to health care (represented by doctors per capita) (Barreca, 2016). By analyzing the remarkable decline of temperature-mortality risks in the USA from 1960 to 2004, Barreca found that the wider use of residential air conditioning may play a major role in reducing heat-related mortality over the 20th century. While the effects of electrification and access to health care on temperature-mortality relationship were not significant. However this paper didn't provided information on the changing MMT values over time, and whether the change in MMTs was due to the wider use of air conditioning.

As suggested by the reviewer, we have revised the paper in the following three points:

- 1) Added the briefly introduce of other mechanisms in addition to physiological acclimatization.
- 2) Weakened the interpretation of GDP.

Details are shown below:

“Humans adapt to climate in several ways, such as physiological, behavioral, and technological adaptations^{8,27-28}. This study reveals that MFT is a good indicator that reflects an important aspect of how humans physiologically adapt to ambient temperature.” (See lines 201-203)

“In addition to physiological acclimatization, the socioeconomic conditions can affect human's behavioral and technological adaptations to ambient temperature. Some studies have confirmed the conclusion^{24,30}. For example, people with low socioeconomic levels are more vulnerable and have a higher mortality risk associated with temperature.” (See lines 211-215)

- 3) Added the references

See lines 398-402.

References:

- [1]. McMichael, A. J., Woodruff, R. E., & Hales, S. Climate change and human health: present and future risks. *The Lancet* **367**, 859-869 (2006).
- [2]. Guo, Y. M., Gasparrini, A., Li, S. S., *et al.* Quantifying excess deaths related to heatwaves under climate change scenarios: A multicountry time series modelling study. *PLoS Med* **15**, e1002629 (2018).
- [3]. Guo, Y. M., Gasparrini, A., Armstrong, B., *et al.* Global variation in the effects of ambient temperature on mortality: a systematic evaluation. *Epidemiology* **25**, 781–789 (2014).
- [4]. Scovronick, N. *et al.* The association between ambient temperature and mortality

- in South Africa: A time-series analysis. *Environ. Res* **161**, 229–235 (2018).
- [5]. Chen, K., Bi, J., Chen, J., *et al.* Influence of heat wave definitions to the added effect of heat waves on daily mortality in Nanjing, China. *Sci Total Environ* **507**, 18–25 (2015).
- [6]. Barreca, A., Clay, K., Deschenes, O., *et al.* Adapting to Climate Change: The Remarkable Decline in the U.S. Temperature-Mortality Relationship over the 20th Century. *Journal of Political Economy* **124**, 105-159 (2016).

REVIEWERS' COMMENTS:

Reviewer #2 (Remarks to the Author):

I appreciate the authors' effort to have responded to my concerns and revised the manuscript. I am satisfied.

One thing which may be to be revised or not is whether to delete the sentence in Line 213: "For example, people with low socioeconomic levels are more vulnerable and have a higher mortality risk associated with temperature." The texts about adaptation through socioeconomic conditions is about the process of how populations can be adapted to ambient temperature. I think Line 213 does not support the previous texts, but rather it merely says the result of the process. It is up to the authors.

Reviewers' comments:

Reviewer #2 (Remarks to the Author):

Comment B1.

I appreciate the authors' effort to have responded to my concerns and revised the manuscript. I am satisfied.

Response B1: Thanks.

Comment B2.

One thing which may be to be revised or not is whether to delete the sentence in Line 213: "For example, people with low socioeconomic levels are more vulnerable and have a higher mortality risk associated with temperature." The texts about adaptation through socioeconomic conditions is about the process of how populations can be adapted to ambient temperature. I think Line 213 does not support the previous texts, but rather it merely says the result of the process. It is up to the authors.

Response B2: We have deleted the sentence. See lines 194-195.